# Trends in the nephrologist workforce in South Africa (2002–2017) and forecasting for 2030

**Dominic Dzamesi Kumashie[1], Ritika Tiwari[2], Muhammed Hassen[1], Usuf M. E. Chikte[2], Mogamat Razeen Davids[1] ***

1 Division of Nephrology, Department of Medicine, Stellenbosch University and Tygerberg Hospital, Cape Town, South Africa, 2 Division of Health Systems and Public Health, Department of Global Health, Stellenbosch University, Cape Town, South Africa

* mrd@sun.ac.za

**Data Availability Statement:** The dataset of demographic data on South African nephrologists cannot be publicly shared as it is small and contains potentially identifying information. Requests for access to this information may be

## Abstract

### Background

The growing global health burden of kidney disease is substantial and the nephrology workforce is critical to managing it. There are concerns that the nephrology workforce appears to be shrinking in many countries. This study analyses trends in South Africa for the period 2002–2017, describes current training capacity and uses this as a basis for forecasting the nephrology workforce for 2030.

### Methods

Data on registered nephrologists for the period 2002 to 2017 was obtained from the Health Professions Council of South Africa and the Colleges of Medicine of South Africa. Training capacity was assessed using data on government-funded posts for nephrologists and nephrology trainees, as well as training post numbers (the latter reflecting potential training capacity). Based on the trends, the gap in the supply of nephrologists was forecast for 2030 based on three targets: reducing the inequalities in provincial nephrologist densities, reducing the gap between public and private sector nephrologist densities, and international benchmarking using the Global Kidney Health Atlas and British Renal Society recommendations.

### Results

The number of nephrologists increased from 53 to 141 (paediatric nephrologists increased from 9 to 22) over the period 2002–2017. The density in 2017 was 2.5 nephrologists per million population (pmp). In 2002, the median age of nephrologists was 46 years (interquartile range (IQR) 39–56 years) and in 2017 the median age was 48 years (IQR 41–56 years). The number of female nephrologists increased from 4 to 43 and the number of Black nephrologists increased from 3 to 24. There have been no nephrologists practising in the North West and Mpumalanga provinces and only one each in Limpopo and the Northern Cape. The current rate of production of nephrologists is eight per year. At this rate, and considering estimates of nephrologists exiting the workforce, there will be 2.6 nephrologists pmp in 2030. There are 17 government-funded nephrology trainee posts while the potential number based on the prescribed trainer-trainee ratio is 72. To increase the nephrologist

submitted to the Health Research Ethics Committee of Stellenbosch University at ethics@sun.ac.za.

**Funding:** The authors received no specific funding for this work but wish to acknowledge the role of the International Society of Nephrology in providing a nephrology training fellowship for DK.

**Competing interests:** The authors have declared that no competing interests exist.

density of all provinces to at least the level of KwaZulu-Natal (2.8 pmp), which has a density closest to the country average, a projected 72 additional nephrologists (six per year) would be needed by 2030. Benchmarking against the 25th centile (5.1 pmp) of upper-middle-income countries (UMICs) reported in the Global Kidney Health Atlas would require the training of an additional eight nephrologists per year.

## Conclusions

South Africa has insufficient nephrologists, especially in the public sector and in certain provinces. A substantial increase in the production of new nephrologists is required. This requires an increase in funded training posts and posts for qualified nephrologists in the public sector. This study has estimated the numbers and distribution of nephrologists needed to address provincial inequalities and achieve realistic nephrologist density targets.

## Introduction

Approximately 10% of the world's population has chronic kidney disease (CKD) and 0.1% has kidney failure [1]. The total number of individuals with kidney disease, including those on kidney replacement therapy (KRT), exceeds 850 million [2]. Over 2 million people die each year due to limited access to KRT, most of whom live in low- and lower-middle-income countries [3]. It is estimated that another 1.7 million die each year from acute kidney injury (AKI) [3, 4].

In Africa, the kidney disease burden is likely to increase substantially over the next decade. Contributing factors include high population growth rates, aging of the population and, by 2040, an increase in the number of people with diabetes of 140% [5, 6]. Although most African patients with kidney failure are unable to access KRT because of insufficient resources, the number of people receiving KRT in Africa is forecast to increase from 0.083 million in 2010 to 0.236 million by 2030 [3]. The most important causes of chronic kidney disease, based on the reported causes of kidney failure in Africans on kidney replacement therapy, are CKD of unknown cause, hypertension and diabetes [5].

It is of great concern that the global nephrology workforce appears to be shrinking despite the growing burden of kidney disease [4]. A shortage in nephrologists and nephrology trainees has been identified across all regions and World Bank income groups, with three-quarters of countries reporting a shortage [7, 8]. The lowest regional density of nephrologists is found in Africa and South Asia (Table 1). Recently, the International Society of Nephrology (ISN) Global Kidney Health Atlas (GKHA) [1] reported that the median density of nephrologists increases with country income; it is 0.2 pmp in low-income countries, 1.6 pmp in lower-middle, 10.8 pmp in upper-middle, and 23.2 pmp in high-income countries. Of the ten countries with the lowest nephrologist density, nine are from the Africa region.

South Africa is an upper-middle-income country (UMIC) with an estimated population in 2017 of 56.5 million people. It had a gross national income (GNI) per capita using the Atlas method (current USD) in 2017 of $5 410 and based on the purchasing power parity (PPP) method (current international USD) of $12 320 [9]. Like many other African countries, South Africa is confronted with a quadruple burden of disease from communicable diseases such as HIV/AIDS and tuberculosis, non-communicable diseases, pregnancy-related diseases, as well as injury and trauma. The estimated population prevalence of CKD in South Africa ranges from 6.1% to 17.3% [10, 11]. The total reported number of patients on KRT in 2017 was 10 744, a prevalence of 190 pmp [12].

**Table 1. Median density of nephrologists and nephrology trainees in International Society of Nephrology (ISN) regions.**

| Region | Number per million population | |
|---|---|---|
| | Nephrologists | Trainees |
| Overall | 8.83 | 1.87 |
| **ISN region** | | |
| Africa | 3.64 | 1.77 |
| Eastern & Central Europe | 16.33 | 3.41 |
| Latin America & the Caribbean | 15.23 | 1.89 |
| Middle East | 6.17 | 1.22 |
| Newly Independent States (NIS) & Russia | 15.68 | 1.64 |
| North America | 24.20 | 1.33 |
| North & East Asia | 12.37 | 3.62 |
| Oceania & Southeast Asia | 3.98 | 0.71 |
| South Asia | 1.17 | 0.41 |
| Western Europe | 21.04 | 3.88 |

Adapted from Osman et al. [7].

Hassen et al. [13] recently reported the density of nephrologists in South Africa to be 2.5 pmp, considerably lower than the upper-middle-income country median of 10.8 pmp [1]. There was a striking sectorial and geographical maldistribution. Most nephrologists were concentrated in the private healthcare sector, a sector that mainly serves the 16% of the population who have private medical insurance. There were no nephrologists in two of the nine provinces (North West and Mpumalanga) and only one each in Limpopo and the Northern Cape [13]. These four provinces are home to one-quarter of the South African population [14].

A situational analysis for workforce planning requires the consideration of factors such as population growth, training capacity, service quality standards, burden of disease, public demand and expectations, organizational efficiency and financing [15]. Countries of various levels of development have differing capacities for collecting and analyzing data and as a result use differing approaches to project future workforce requirements [15]. Four main methods are used: (i) the workforce-to-population ratio method, (ii) the health needs method, which considers burden of disease and the workforce required to respond, (iii) the service demands method, which considers health services standards and utilization rates and (iv) the services target method, which specifies production targets for health services [15].

We have utilized the workforce-to-population ratio method as well as aspects of the other three approaches. In this report, we describe trends in nephrologist numbers, demographic profile and geographical distribution for the period 2002–2017. We also assess the current training capacity, the rate of production of new nephrologists and the rate of exit of nephrologists from the workforce. Estimates of the need for nephrologists and of the gap between the current trend and the desired targets are used to forecast the changes required to meet these targets by 2030.

## Methods

### A. Trends in the numbers and demographic profile of nephrologists, current training capacity and the rate of production of new nephrologists

Data on the numbers and demographic profile of nephrologists in South Africa were obtained from the Health Professions Council of South Africa (HPCSA) and cross referenced with data from the Colleges of Medicine of South Africa (CMSA) as well as relevant published literature.

The HPCSA is the regulatory body which registers health practitioners and the CMSA is the national examination authority for specialists in. The data spanned a 16-year period from 2002 to 2017. The variables captured included age, sex, population group, nationality and provincial location.

Training capacity was assessed by determining the numbers of government-funded posts for consultant nephrologists and nephrology trainees ("senior registrars") at training institutions and their affiliated hospitals, as well as the number of HPCSA training post numbers for nephrology at each training institution. This information was supplied by the heads of the academic divisions of nephrology at these institutions.

The rate of production of new nephrologists was assessed using data from the CMSA on successful candidates in the nephrology certificate examinations and data from the HPCSA on new nephrologist registrations.

## B. Nephrologist workforce forecasting 2018–2030

Our forecasting considered the need for nephrologists, the rate of production of new nephrologists, the exit of nephrologists from the workforce, and estimation of the gap between the current trend and the desired targets.

**Forecasting the need for nephrologists.** We considered three approaches to estimating the need for nephrologists and setting targets for 2030: reducing inequities in provincial nephrologist densities, reducing the gap between public and private sector nephrologist densities and international benchmarking.

*i. Reducing inter-provincial inequalities*. The need for nephrologists was estimated based on reducing the inter-provincial inequalities in nephrologist density. The calculations were based on raising the density in the most underserved provinces to that of KwaZulu-Natal (2.8 pmp), which is the province with a nephrologist density closest to the national average of 2.5 pmp [13].

*ii. Bridging the public-private sector gap*. Data from Statistics South Africa [14] and the Council of Medical Schemes [16] were used to determine the proportion of the population accessing the public and private healthcare sectors. These data were used as denominators in determining the nephrologist density in the two sectors. We then forecast the number of nephrologists needed to address the private-public disparity.

*iii. Using international benchmarking*. For estimating the overall need for nephrologists, we used data on nephrologist density in UMICs from the Global Kidney Health Atlas [1, 7] for benchmarking and also considered the British Renal Society (BRS) recommendations of one nephrologist for every 75 patients on KRT [17]. The BRS recommendations use the number of patients on KRT as an indicator of the overall workload of a nephrologist.

**Forecasting the production of new nephrologists.** Based on the historical trends, and especially the more recent period of 2010–2017, the average annual supply of new nephrologists was used to project the growth to the year 2030.

**Forecasting the exit of nephrologists from the workforce.** We considered nephrologists exiting the workforce on account of retirement, death/disability and migration. Hassen et al. [13] noted that a significant proportion of South African nephrologists (12.5% in the private sector and 8% in the public sector) indicated their desire to retire before the age of 65 years, which is the statutory retirement age in the public service. On the other hand, there was a significant number of nephrologists still working who were over the age of 65 (11% of the workforce). Working beyond 65 years is more likely in the case of private sector nephrologists and will mitigate, in part, the effect of those who go on early retirement. For our calculations, we have assumed the retirement age to be 65 years.

**Gap estimation.**   For our projections, we estimated the workforce for each year as the existing number of active nephrologists plus the supply of new nephrologists less those exiting. The gap to be addressed by 2030 is the difference between the forecasted number of nephrologists if current trends are maintained and the targets identified through examining inter-provincial inequities, public-private sector comparisons and international benchmarking.

**Ethical considerations.**   Ethical approval for the study was obtained from the Stellenbosch University Health Research Ethics Committee (reference number S16/05/094). A waiver of individual informed consent was granted.

## Results

### A. Trends in the demographic profile of nephrologists, current training capacity and the rate of production of new nephrologists

The number of nephrologists nearly tripled from the year 2002 (n = 53) to 2017 (n = 141), with an average annual increase of 6% over the period. The density in 2017 was 2.5 pmp. The average annual increase in the number of nephrologists was 3% between 2002 and 2006, 6.5% between 2007 and 2011, and 7% between 2012 and 2017. The number of paediatric nephrologists increased from 9 in 2002 to 22 in 2017.

In 2002, 62% of nephrologists were aged <50 years and 18% were 60 years or older. In 2017, 55% (n = 78) were aged <50 years and 20% (n = 28) were 60 years or older, with 11% being older than 65 years. In 2002, the median age was 46 years [interquartile range (IQR) 39–56 years] whereas in 2017 it was 48 years (IQR 41–56 years).

In 2017, most nephrologists (n = 98, 70%) were male. The proportion of female nephrologists increased from 8% (n = 4) in 2002 to 30% (n = 43) at the end of 2017 (Fig 1).

Regarding population group, most nephrologists were White or Indian, with these groups representing 89% of the total in 2002 and 77% in 2017. The proportion of Black nephrologists increased from 6% (n = 3) in 2002 to 17% (n = 24) in 2017. The proportion of nephrologists of mixed ancestry ("Coloured") remained at 6% (3 nephrologists in 2002 and 9 in 2017).

The province of Gauteng, which is the most populous and home to one quarter of the South African population, has the highest proportion of nephrologists (41%), followed by the Western Cape (24%), KwaZulu-Natal (21%), Free State (5%) and Eastern Cape (4%). Over the study period, the relative distribution of nephrologists across the different provinces of South Africa has been approximately the same. There have been no nephrologists practising in the North West and Mpumalanga provinces and only one each in Limpopo and the Northern Cape.

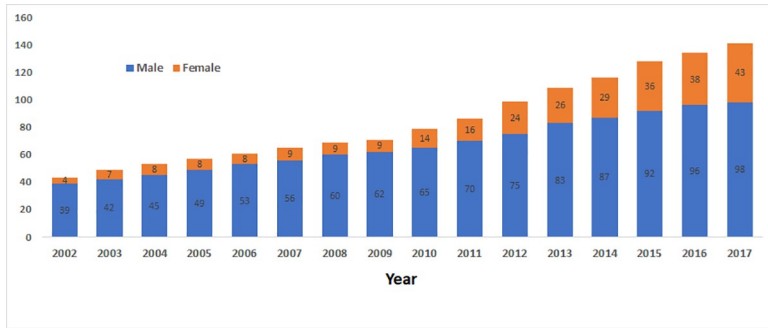

**Fig 1. Trends in the number and sex distribution of nephrologists, 2002 to 2017.**

**Table 2. Training capacity for adult and paediatric nephrologists in South Africa.**

| Institution | Nephrologist posts | | Funded training posts | | HPCSA training numbers | |
|---|---|---|---|---|---|---|
| | Adult | Paeds | Adult | Paeds | Adult | Paeds |
| Stellenbosch University | 3 | 1 | 1 | 1 | 8 | 3 |
| University of Cape Town | 4 | 3 | 2 | 0[a] | 8 | 4 |
| University of KwaZulu-Natal | 4 | 2 | 4[b] | 0[c] | 10 | 2 |
| University of Pretoria | 3 | 1 | 1 | 0 | 4 | 3 |
| University of the Free State | 2 | 0 | 1[d] | 0 | 3 | 0 |
| University of the Witwatersrand | 9 | 4 | 7 | 0[e] | 14 | 6 |
| **Totals** | **25** | **11** | **16** | **1** | **47** | **18** |

Government-funded posts for nephrologists and nephrology trainees, and HPCSA training post numbers per institution. Data as at January 2020.

[a] A training post was sacrificed to create the third paediatric nephrologist post.

[b] These are four specialist physician posts being used for training.

[c] A specialist physician post is used for training, when available.

[d] This is a specialist physician post being used for training.

[e] A specialist paediatrician post is used for training, when available.

The training of nephrologists in South Africa takes place at six universities and their associated public sector teaching hospitals. Table 2 shows the training centres with nephrologist numbers, the number of government-funded nephrology trainee posts (usually "senior registrar" posts) and the HPCSA training post numbers per institution, reflecting training capacity. The ratio of nephrologists to trainees is determined by the HPCSA and is currently 1:2.

Between 2002 and 2017, a total of 85 South Africans completed their sub-specialty training and registered as nephrologists. Fig 2 shows the annual number of trainees passing the CMSA Certificate in Nephrology examinations and the new nephrologist registrations with the HPCSA. Over the 16-year period (2002–2017), an average of 5 nephrologists were added to the workforce annually, with an average of 2 added annually between 2002 and 2009, and 8 added annually between 2010 and 2017.

During the same period (2002–2017), 63 doctors from African countries other than South Africa have been trained as nephrologists, funded mostly through the ISN Fellowship Program

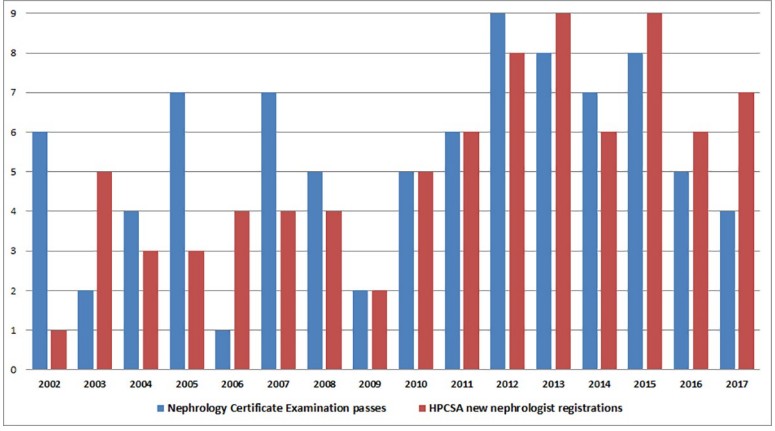

**Fig 2. Annual number of trainees passing the colleges of medicine of South Africa certificate in nephrology examination and number of new HPCSA nephrologist registrations.** Abbreviation: HPCSA, Health Professions Council of South Africa.

[18]. Nigeria (n = 15) and Kenya (14) have had the highest number of trainees. The University of Cape Town, University of the Witwatersrand and Stellenbosch University have been responsible for most of this training.

## B. Nephrologist workforce forecasting 2018–2030

**Forecasting the need for nephrologists.** *i. Reducing inter-provincial inequalities.* Table 3 shows nephrologist density by province, ranked in decreasing order. There is wide variation in the distribution of nephrologists, with the Western Cape and Gauteng the highest ranked according to this metric (5.5 and 4.3 pmp, respectively). Considering that the national density of nephrologists is 2.5 pmp [13], a reasonable target would be to raise the nephrologist density in all provinces to the level of KwaZulu-Natal (2.8 pmp), which is closest to the national average.

*ii. Bridging the public-private sector gap.* Most nephrologists (55.3%, n = 78) are working in the private sector, serving the 16% of the population who are medically insured, while the remainder (44.7%, n = 63) serve the 84% of the population reliant on the public sector. This translates into a density of 8.5 nephrologists pmp in the private sector and 1.3 pmp in the public sector. The private sector nephrologist density is comparable to the median density in UMICs (10.8 pmp) [1]. To address this private/public inequity, the public sector would currently have to be served by 414 nephrologists instead of 63. This shortfall of 351 nephrologists is projected to increase to 401 in 2030, when the national population is expected to reach 65 million [19], assuming that the same proportions of the population would be accessing the public and private sectors at that time.

*iii. International benchmarking.* According to the GKHA 2019 report, the 50th and 25th centile of nephrologist density for UMICs are 10.8 pmp and 5.1 pmp, respectively [1]. In 2030, South Africa would need to have 702 nephrologists or 332 nephrologists to reach the 50th or 25th centile targets, respectively (Fig 3).

Based on the BRS recommendation of one nephrologist per 75 patients on KRT, the overall number of nephrologists in South Africa could be regarded as adequate. There were 10 744 patients on KRT in 2017 [12] and 141 nephrologists, yielding a ratio of 1:76. However, KRT in the South African public sector is severely constrained by limited resources and therefore the size of the KRT population is an unreliable proxy for the total nephrology workload. If KRT were freely available in the public sector, with the same prevalence as in the private sector (855 pmp), there would be 48 325 patients on KRT in the country and a shortfall of 503 nephrologists, using this approach.

**Forecasting the production of new nephrologists.** Eight nephrologists are produced annually, meaning that 104 new nephrologists would be added to the nephrology workforce

**Table 3. Density of nephrologists per million population, by province, in 2017.**

| Province | Population | Number of nephrologists | Nephrologists per million population (pmp) | Rank |
|---|---|---|---|---|
| Western Cape | 6510300 | 36 | 5.5 | 1 |
| Gauteng | 14278700 | 61 | 4.3 | 2 |
| KwaZulu-Natal | 11074800 | 31 | 2.8 | 3 |
| Free State | 2866700 | 6 | 2.1 | 4 |
| Eastern Cape | 6498700 | 6 | 0.9 | 5 |
| Northern Cape | 1225555 | 1 | 0.8 | 6 |
| North West | 3856200 | 0 | 0.0 | 7 |
| Mpumalanga | 4444200 | 0 | 0.0 | 8 |
| Limpopo | 5778400 | 0 | 0.0 | 9 |

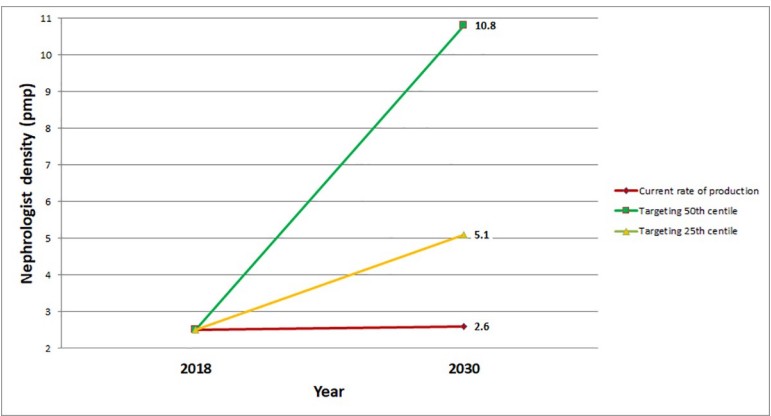

**Fig 3. Benchmarking South African nephrologist density against upper-middle-income country targets.** Based on the 50[th] and 25[th] centile densities from the Global Kidney Health Atlas [1]. Abbreviation: pmp, per million population.

from 2018 to 2030, if current trends are maintained. This rate of production is limited by the number of funded training posts nationally (currently 17). Based on the current number of 36 nephrologists at the training institutions, the HPCSA training post numbers could be increased to 72. This reflects potential training capacity; if sufficient funding for these posts were available, 72 new nephrologists could be produced every two years.

**Forecasting the exit of nephrologists from the workforce.** Between 2002 and 2017, 11 nephrologists were lost through death and emigration. Extrapolating this rate of attrition to the period 2018 to 2030, an assumption of a loss of 9 nephrologists could be made and, with an additional 62 nephrologists expected to be lost through retirement at age 65 by 2030, this brings the total projected loss of nephrologists by 2030 to 71.

**Gap estimation.** South Africa's population is expected to increase to 65 million by 2030 [19]. If the current rate of production of new nephrologists is maintained and considering our estimates of nephrologists exiting the workforce, the number of active nephrologists in South Africa in 2030 is projected to be 174, a density of 2.7 pmp. The gaps in the nephrologist work-force that would need to be addressed to achieve selected targets by 2030 are as follows:

i. Reducing inter-provincial inequalities: to raise the nephrologist density of the lowest six provinces to the current level of KwaZulu-Natal (2.8 pmp), a total of 55 additional nephrologists are currently needed. The Free State and Northern Cape need 2 nephrologists each, the North West needs 11, Mpumalanga and the Eastern Cape need 12 each and Limpopo needs 16 nephrologists. Projecting to 2030, and assuming that the relative provincial population distribution is maintained, 72 additional nephrologists would be needed (KwaZulu-Natal would need 5, Free State and Northern Cape would need 3 each, North West would need 12, Mpumalanga and the Eastern Cape would need 15 each and Limpopo would need 19).

ii. Eliminating public-private sector inequalities: to match the current nephrologist density in the private sector (8.5 pmp), the public sector would currently need 351 additional nephrologists and 401 additional nephrologists would need to be trained by 2030.

iii. International benchmarking: to reach the 25th centile of nephrologist density in UMICs (5.1 pmp), 191 new nephrologists must be produced by 2030, a rate of 16 nephrologists per year, resulting in a total of 332 nephrologists.

## Discussion

Our study has revealed several important findings. Although the number of nephrologists has nearly tripled since 2002, the overall density is only 2.5 pmp, well below the median for UMICs. At the current rate of production (8 nephrologists per annum) and with 71 nephrologists expected to exit the workforce in the next 12 years, the nephrologist density in 2030 is forecasted to be 2.6 pmp (174 nephrologists). Given the increasing burden of disease, this is clearly insufficient to meet the demand for specialist nephrology services.

The demographic profile of the workforce is changing, with the proportion of female nephrologists increasing from 8% to 30%, and Black nephrologists from 6% to 17%.

The distribution of nephrologists by healthcare sector is markedly unequal, with a density of 8.5 pmp in the private sector and 1.3 pmp in the public sector. The provincial distribution is also unequal, with the Western Cape (5.4 pmp) and Gauteng (4.2 pmp) having the highest nephrologist densities. There have been no registered nephrologists in the North West and Mpumalanga provinces and only one each in Limpopo and the Northern Cape.

Training is currently operating well below the potential capacity, based on the numbers of consultant nephrologists and the allocated HPCSA training post numbers at the training institutions. Actual training capacity is dependent on government-funded trainee ("senior registrar") posts and these are grossly inadequate, setting the limit on the production of new nephrologists at approximately 8 per annum. If adequate funding for training posts were available, 72 new nephrologists could be produced every 2 years.

We considered two targets as being realistic for medium-term workforce planning to 2030: (i) 72 additional nephrologists would need to be trained to reduce the inter-provincial disparities so that all provinces are at least at the nephrologist density of KwaZulu-Natal (2.8 pmp) and (ii) 191 additional nephrologists would need to be trained to reach the 25th centile for nephrologist density in UMICs (5.1 pmp).

To address inter-provincial inequities and to reach the UMIC 25th centile by 2030 would require producing 14 and 16 nephrologists per annum, respectively, instead of the current 8 per annum.

At the current rate of production, the country will not have the nephrologists needed to address the increasing burden of kidney disease. This will lead to a further increase in workload for the active nephrologists which may result in burnout, early retirements and compromised quality of care for patients with kidney disease [13]. This scenario may also lead to a decreased interest in nephrology as a career, a problem that has been identified as a key factor in the dwindling nephrology workforce in the United States.

The lack of posts in the public sector for newly qualified nephrologists fuels the maldistribution between the private and public sectors as well as between urban and rural areas. The inability to retain nephrologists in the public sector (which is responsible for all the training of nephrologists) does not augur well for the future of the nephrology workforce in the public sector and, by extension, for our training capacity.

Since 2004, a "rural allowance" has been implemented to attract and retain healthcare workers in designated hospitals in underserved, rural areas. For medical specialists like nephrologists, this allowance amounts to 18–22% of their basic salary. Heads of provincial health departments may also include non-rural "inhospitable health institutions" under this agreement [20, 21]. This intervention has been considered to be partially effective, with some weaknesses in its implementation and some unintended negative consequences [20]. Nephrology services which include kidney replacement therapy are often based at larger hospitals and these hospitals may not covered by this agreement.

There is a need for urgent planning by government and other key stakeholders to increase the number of funded training posts and to create posts to employ more nephrologists in the public healthcare sector. Government and training institutions should explore partnerships with the corporate sector to augment funds for training. In the short term, the retirement age for public sector nephrologists could be increased or retired nephrologists could be retained on a part-time basis to assist with service delivery, training and research.

Hassen et al. [13] reported that the provincial maldistribution of nephrologists was mainly due to individual choices based on the demands of family life, with additional considerations being high workload, transportation difficulties and limited access to resources for kidney care. Some nephrologists who were working in under-resourced provinces did so due to contractual obligations towards those who had sponsored their training. Such policies may increase the numbers of nephrologists settling in under-served provinces and should be more widely adopted [13]. Nephrologists in these areas should act as positive role models and mentors for younger colleagues, to encourage them to pursue a career in nephrology and to practice in under-served areas after the completion of their training [13].

## Limitations

Our study focused only on nephrologists and not on the other healthcare professionals in the multi-disciplinary nephrology team. Studies to investigate human resources issues relevant to these colleagues are needed to ensure that comprehensive kidney care can be delivered. A more detailed examination of paediatric nephrology human resources is also required. This study has also not dealt with the expansion of infrastructure and other supportive services which would be essential in improving kidney care in South Africa.

The forecasting in this study is based mainly on workforce-to-population ratio analysis. This method is the least demanding in terms of data but does not consider possible changes in key variables such as burden of disease, service delivery models, facility planning and workloads, skills mix, remuneration systems or health worker productivity. The workforce-to-population ratio approach also assumes that all physicians are equally productive and will remain so while active.

## Conclusions

Deliberate and detailed nephrology workforce planning is critical to addressing the growing burden of kidney disease and to improve the quality of kidney care in South Africa. The country has insufficient nephrologists, especially in the public sector and in certain provinces. A substantial increase in the production of new nephrologists is required. This requires an increase in funded training posts and posts for qualified nephrologists in the public sector. Should these actions be taken, there is a realistic chance of successfully addressing provincial inequalities and achieving nephrologist densities comparable to other UMICs.

## Supporting information

**S1 File. Numbers of South African nephrologists passing the certification examination and registering with the HPCSA.**
(XLS)

## Author Contributions

**Conceptualization:** Usuf M. E. Chikte, Mogamat Razeen Davids.

**Data curation:** Dominic Dzamesi Kumashie, Ritika Tiwari, Muhammed Hassen, Mogamat Razeen Davids.

**Formal analysis:** Dominic Dzamesi Kumashie, Ritika Tiwari.

**Supervision:** Usuf M. E. Chikte, Mogamat Razeen Davids.

**Writing – original draft:** Dominic Dzamesi Kumashie, Mogamat Razeen Davids.

**Writing – review & editing:** Dominic Dzamesi Kumashie, Ritika Tiwari, Muhammed Hassen, Usuf M. E. Chikte, Mogamat Razeen Davids.

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
