## [Decision Letter · Decision Letter 0]

4 May 2021

PONE-D-21-07013

Trends in the nephrologist workforce in South Africa (2002–2017) and forecasting for 2030

PLOS ONE

Dear Prof Mogamat Razeen Davids

Thank you for submitting your manuscript to PLOS ONE. After careful consideration, we feel that it has merit but two concerns raised by the reviewers need to be addressed before the paper can be considered for publication. Therefore, we invite you to submit a revised version of the manuscript that addresses the points raised during the review process.

We look forward to receiving your revised manuscript.

Kind regards,

Rajendra Bhimma, PhD

Academic Editor

PLOS ONE

Journal Requirements:

3. In your ethics statement in the manuscript and in the online submission form, please ensure that you have discussed whether all data/samples were fully anonymized before you accessed them and/or whether the IRB or ethics committee waived the requirement for informed consent.

"This study was conducted during the ISN Fellowship of Dominic Kumashie. We

gratefully acknowledge the support of the ISN."

 "The authors received no specific funding for this work."

Additional Editor Comments:

Please see comments by reviewers.

Reviewer 1

Suggestion: Include statistics from sub-Saharan Africa on the projected increase in the prevalence of chronic kidney disease over the next decade as well as the main causes of chronic kidney disease in the region.

Reviewer 2

Do the authors feel that Nephrologists should also be 'induced' to work in areas currently understaffed, with specific 'inducement allowances' to attract them to those areas?

Reviewers' comments:

Reviewer's Responses to Questions

**Comments to the Author**

1. Is the manuscript technically sound, and do the data support the conclusions?

Reviewer #1: Yes

Reviewer #2: Yes

2. Has the statistical analysis been performed appropriately and rigorously? 

Reviewer #1: Yes

Reviewer #2: I Don't Know

3. Have the authors made all data underlying the findings in their manuscript fully available?

Reviewer #1: Yes

Reviewer #2: Yes

4. Is the manuscript presented in an intelligible fashion and written in standard English?

Reviewer #1: Yes

Reviewer #2: Yes

5. Review Comments to the Author

Reviewer #1: Abstract and introduction:

Both were clear with well defined aims of the study. The main research question was distinct. Appropriate use of supporting literature to relate to main research question.

Suggestion: Include statistics from sub-Saharan Africa on the projected increase in the prevalence of chronic kidney disease over the next decade as well as the main causes of chronic kidney disease in the region.

Method:

Would be reproducible. It is concisely written.

Figures and tables are concise and accurate and correctly presented. Appropriate to answer the research question and conforms to ethical guidelines. Enough data was obtained to draw conclusions. Limitations mentioned were appropriate.

Results were correlated with the tables and figures and support conclusions. Statistical analysis was acceptable. Results provided can be reproduced.

Discussion highlighted critical points relevant to the study.

Conclusions:

Answered what the authors had put forward in the introduction.

Overall the writing quality and clarity was suitable.

Reviewer #2: Beautiful paper highlighting very important workforce data present & projected in South Africa. Do the authors feel that Nephrologists should also be 'induced' to work in areas currently understaffed, with specific 'inducement allowances' to attract them to those areas?

6. PLOS authors have the option to publish the peer review history of their article (what does this mean?). If published, this will include your full peer review and any attached files.

Reviewer #1: No

Reviewer #2: No

---

## [Author Response · Author response to Decision Letter 0]

7 Jun 2021

Please see the "Response to reviewers" document attached.

---

## [Decision Letter · Decision Letter 1]

27 Jul 2021

Trends in the nephrologist workforce in South Africa (2002–2017) and forecasting for 2030

PONE-D-21-07013R1

Dear Dr. Davids,

We’re pleased to inform you that your manuscript has been judged scientifically suitable for publication and will be formally accepted for publication once it meets all outstanding technical requirements.

Kind regards,

Rajendra Bhimma, PhD

Academic Editor

PLOS ONE

Additional Editor Comments (optional):

Dear Prof R Davids

Thank you for the submission of your manuscript to PLOS ONE. I have been through all the responses and I am satisfied that these have been adequately addressed. The manuscript has been forwarded to the Editor-in-Chief for a final decision.

Reviewers' comments:

Reviewer's Responses to Questions

**Comments to the Author**

1. If the authors have adequately addressed your comments raised in a previous round of review and you feel that this manuscript is now acceptable for publication, you may indicate that here to bypass the “Comments to the Author” section, enter your conflict of interest statement in the “Confidential to Editor” section, and submit your "Accept" recommendation.

Reviewer #1: All comments have been addressed

2. Is the manuscript technically sound, and do the data support the conclusions?

Reviewer #1: Yes

3. Has the statistical analysis been performed appropriately and rigorously? 

Reviewer #1: Yes

4. Have the authors made all data underlying the findings in their manuscript fully available?

Reviewer #1: Yes

5. Is the manuscript presented in an intelligible fashion and written in standard English?

Reviewer #1: Yes

6. Review Comments to the Author

Reviewer #1: Minor revisions have been adequately addressed. The requested review concerning the anticipated increase in patients with CKD and main causes have been mentioned in the revision.

7. PLOS authors have the option to publish the peer review history of their article (what does this mean?). If published, this will include your full peer review and any attached files.

Reviewer #1: No

---

## [Editor Report · Acceptance letter]

3 Aug 2021

PONE-D-21-07013R1 

Trends in the nephrologist workforce in South Africa (2002–2017) and forecasting for 2030 

Dear Dr. Davids:

I'm pleased to inform you that your manuscript has been deemed suitable for publication in PLOS ONE. Congratulations! Your manuscript is now with our production department. 

Kind regards, 

on behalf of

Professor Rajendra Bhimma 

Academic Editor

PLOS ONE